# When weight is an encumbrance; avoidance of stairs by different demographic groups

**Frank F. Eves** ⬦ *

School of Sport, Exercise and Rehabilitation Sciences, University of Birmingham, Birmingham, England, United Kingdom

* F.F.Eves@bham.ac.uk

## Abstract

### Background

Locomotion is an energy costly behaviour, particularly when it entails raising weight against gravity. Minimization of locomotor costs appears a universal default. Avoidance of stair climbing helps humans minimise their energetic costs. In public access settings, demographic subgroups that raise more 'dead' weight than their comparison groups when climbing are more likely to avoid stairs by choosing the escalator. Individuals who minimise stair costs at work, however, can accumulate a deficit in energy expenditure in daily life with potential implications for weight gain. This paper tests the generality of avoidance of stairs in pedestrians encumbered by additional weight in three studies.

### Methods

Pedestrian choices for stairs or the alternative were audited by trained observers who coded weight status, presence of large bags and sex for each pedestrian. Sex-specific silhouettes for BMIs of 25 facilitated coding of weight status. Choices between stairs and a lift to ascend and descend were coded in seven buildings (n = 26,981) and at an outdoor city centre site with the same alternatives (n = 7,433). A further study audited choices to ascend when the alternative to stairs was a sloped ramp in two locations (n = 16,297). Analyses employed bootstrapped logistic regression (1000 samples).

### Results

At work and the city centre site, the overweight, those carrying a large bag and females avoided both stair climbing and descent more frequently than their comparison groups. The final study revealed greater avoidance of stairs in these demographic subgroups when the alternative means of ascent was a sloped ramp.

### Discussion

Minimization of the physiological costs of transport-related walking biases behaviour towards avoidance of stair usage when an alternative is available. Weight carried is an encumbrance that can deter stair usage during daily life. This minimization of physical

**Data Availability Statement:** All files are available at https://doi.org/10.5061/dryad.w0vt4b8m5.

**Funding:** FFE G0802070/91321 Medical Research Council, UK The funders had no role in study design, data colection and analysis, decision to publish, or preparation of the manuscript.

**Competing interests:** The author has declared that no competing interests exist.

activity costs runs counter to public health initiatives to increase activity to improve population health.

## Introduction

Use of stairs, instead of escalators or lifts, is one of a range of transport-related behaviours that allow accumulation of incidental physical activity during daily life. Although minimum bouts of 10 minutes duration were recommended for cardio-respiratory benefits, the most recent evidence review concluded that short duration bouts were beneficial [1]. For body weight, all physical activity requires energy expenditure that counters intake that can lead to weight gain. Stair climbing is an energy demanding behaviour that involves raising body weight against gravity; it requires 9.6 times the energy expenditure of rest outside the laboratory [2]. An 80 kg individual going upstairs in their own home ten extra times each day for a year expends the energy equivalent to three pounds of fat [3]. Repeated for ten years, expenditure equivalent to over two stone could be accumulated. Conversely, an individual avoiding climbing ten, 3m flights of stairs each day could accumulate reduced expenditure relative to intake. Weight gain has many potential drivers; this paper summarises data for one of them, avoidance of energy expenditure as part of daily life by avoiding stair climbing.

Although interventions to increase stair climbing are part of current public health strategy [4–7], the primary goal of seminal research on stair climbing was an '*unobtrusive measure of physical activity in natural settings*' (page 1540 [8]). Auditing of choice between stairs and escalators unobtrusively assessed physical activity choices in overweight and obese individuals in a shopping mall, an airport, as well as bus and train stations. Both Meyers et al., (1980) and Brownell et al., (1980) reported less frequent stair climbing in overweight than healthy weight pedestrians. A recent review of public access settings confirms these reports; all nine studies with relevant data report significantly less frequent stair climbing in overweight pedestrians [9]. Consistently, overweight pedestrians, offered the opportunity to avoid energy expenditure, do so more frequently than healthy weight individuals. What the data also reveal is that *most* healthy weight pedestrians avoid the stairs, 91.0% in the original research by Brownell and colleagues (n = 47,548 [8]) and 92.4% prior to intervention in 15 different shopping mall studies (n = 355,069 [9]). While avoidance of energy expenditure on stairs is more frequent in the overweight, it is common to all pedestrians, including healthy weight ones. This prevalence of avoidance reflects a bias to minimise the energetic costs of locomotion (see general discussion).

To climb stairs, an individual must raise their body weight against gravity. Raising weight when climbing stairs entails two and a half times the energetic costs of purposeful walking [2]. Any additional weight carried will increase the energetic cost. Individuals carrying large bags and females also avoid stairs more than their comparison groups [9]. An average woman has a greater percentage of her weight as body fat (25%) than an average man (12.5%); she would raise more 'dead' weight against gravity for the same set of stairs [10]. Climbing is more energetically taxing on the resources of overweight pedestrians, those carrying large bags and females because of the 'dead' weight they must carry upwards.

### Avoidance of stairs at work

While avoidance of stair climbing has consequences for energy balance, a choice between stairs and escalators in a public access setting such as a station is a relatively infrequent daily event.

At work, however, stairs and their alternative the lift are a more frequently encountered choice. An individual at work who avoids stairs more than a colleague could accumulate a disparity in energy expenditure as part of daily life with obvious potential consequences for individual differences in weight gain. Employed individuals spend half of their waking life at work [11]. As a result, choice behaviour at work may better address the generality of the avoidance posed by the original researchers of stair choice. This paper provides data on the effects of demographic grouping on avoidance of stair climbing initially at work and then in public access settings, to test for the generality of the effects of demographics on avoidance.

## Preliminary study

Table 1 summarises the results of a preliminary literature review on previously reported significant effects of the demographics of weight status, carrying a large bag and being female on choice between stairs and lifts at work (S1 File; Effects of demographic grouping on workplace stair avoidance).

As can be seen from Table 1, the available data were sparse. The two studies testing effects of weight status, and three of the five recording the presence of large bags, reported greater avoidance of stairs in overweight and encumbered pedestrians respectively, a pattern consistent with greater avoidance by these demographic groups in public access settings. Effects for sex on avoidance were mixed. Nine reported that females avoided stairs more than males, five reported the opposite disparity and seven no significant difference. Analysis of this distribution revealed that females at work did not avoid stairs significantly more than males (binomial test p = .808). As reported in S1 File, there were no obvious co-occurrences of sex differences in avoidance with measurement or analysis choices that might explain the mixed evidence. The anomalies of greater avoidance in men might reflect differing locations by gender of meaningful journey ends within the building. For comparison, the significant effects of demographic grouping in public access studies [9] revealed that females avoided stairs more than males in 32 out of 42 studies ($p = .0005$), the overweight more than the healthy weight (n = 9/9, $p = .002$), and those carrying large bags more than the unencumbered (n = 14/17, $p = .006$).

One further point from the additional material is informative. The percentage choosing the lift at baseline was included in the table to illustrate the much less frequent avoidance at work than in public access settings [12]. The sample size weighted average of the avoidance, 61.0%, meant that choosing to expend energy climbing at work was more than five times more frequent than it was in shopping malls (39.0%; n = 181,168 vs. 7.6%; n = 355,069 [9]). Biases against expenditure were reduced at work. Provision of a lift to ascend, as opposed to an escalator in a mall, is the most plausible explanation for this discrepancy between contexts [12,13]. In a public access setting, choice of either the stairs or an adjacent escalator typically incurs a minimal time penalty. At work, however, a pedestrian will have to wait for any lift not at their floor, slowing the journey [13]. Time is important to pedestrians in public access settings [13–15] and at work [16,17]. When the mechanised alternative is a lift, the stairs may provide a

**Table 1. Summary of previous significant effects in workplaces of demographics on avoidance of stair climbing.**

| | Demographic (number of studies) | | | | | | | | |
|---|---|---|---|---|---|---|---|---|---|
| | Weight Status (n = 2) | | | Carrying large bag (n = 5) | | | Sex (n = 21) | | |
| | OW >[a] HW | = | HW > OW | Bag > No bag | = | No Bag > Bag | F > M | = | M > F |
| n = | 2 | 0 | 0 | 3 | 2 | 0 | 9 | 7 | 5 |

[a] > indicates that one group avoided stairs significantly more than the other, e.g. F > M means females avoided stairs significantly more than males. = indicates no significant differences between the demographic groups for the studies in that column, OW = overweight, HW = healthy weight, F = females, M = males.

quicker option. More available lifts in a workplace, however, will reduce waiting times and stair avoidance is increased as a result [13,18]. Multivariate analyses of the first study formally test effects of lift availability and a second structural determinant, direction of travel, independent of the effects of demographic grouping.

At 9.6 metabolic equivalents (METs) of the resting state, stair climbing is a vigorous activity that requires twice the energy of stair descent (4.9 METs [2]). Direction of travel on the stairs is relevant to energy costs that might be avoided. If avoidance is to generalise, however, one would expect effects of demographic grouping for descent as well as ascent; walking down stairs costs more than standing in a lift. In public access settings, only two studies provide data. Meyer et al., (1980) reported more frequent avoidance of stair descent for obese than healthy weight pedestrians [8] whereas Webb and Eves (2007) reported greater avoidance in females and those carrying large bags [19].

## Study 1

Given the paucity of available data on demographics at work, and the mixed data for sex, the first study reports data from workplace observations. The data test for the generality of demographic effects on avoidance by coding, and separately analysing, stair ascent and descent. Based on the sparse public access data, demographic differences were predicted for both directions of travel and less avoidance was predicted for the lower energetic cost of descending.

### Methods

Ethical approval for the studies was obtained from the University of Birmingham ethics sub-committee. The ethics committee did not require the consent of observed pedestrians. Approval to observe the employees in the first study was obtained from the management of the firms.

Inconspicuous observers monitored stair and lift choices (up n = 14,607; down n = 12,334) from 9:00am to 4:00pm at the ground floor in seven separate buildings for 8–12 days. (The number of observations, lifts and floors in each building and the percentage stair use are summarised in S2 File). Following training, observers coded choice, direction of travel, gender and presence of large bags/boxes using previously employed criteria [20]; large bags were sizes greater than a briefcase or small rucksack. The silhouettes for male and female BMIs of 25 were watermarks on the coding sheet to facilitate coding of weight status (c.f. [3,21]). Multiple coding of lift and stair choices revealed excellent Kappas [κ] for method of ascent (κ = 0.99), weight status (κ = 0.94), presence of large bags (κ = 0.92) and sex (κ = 0.99).

### Analyses

The tabulated data contain percentage avoidance with 95% confidence intervals (CIs) to facilitate inspection; non-overlapping CIs indicate differences between the compared groups and conditions. Formal analyses employed logistic regression with bootstrapping (1000 samples) to control for the potential non-independence of observations at work. Stair/lift choice was the dichotomous dependent variable and the potential predictor variables were direction of travel, weight status and sex. A second structural aspect, number of lifts, was also included as more frequent avoidance occurs as lift availability increases [13,18]. Buildings with one lift were compared with those providing two. Although there are also effects of number of floors on stair use (e.g. 18]), taller buildings typically have more lifts to accommodate the greater number of employees. As a result, number of lifts and floors co-occur and inclusion of both variables in a relatively small data set of different buildings was precluded by this multi-collinearity.

**Table 2. Percentage avoidance of stairs (95% CI)[a] by lift number, direction of travel and demographic group.**

| Demographic | Total n = | One lift | | Total n = | Two lifts | |
| --- | --- | --- | --- | --- | --- | --- |
| | | Up | Down | | Up | Down |
| Overweight | 2,067 | 49.2 (45.9,52.4) | 22.3 (19.1,25.6) | 3,973 | 53.4 (50.9,55.7) | 34.7 (32.2,37.3) |
| Healthy weight | 3,873 | 28.8 (26.6,31.1) | 13.1 (11.3,15.0) | 17,068 | 49.1 (47.9,50.3) | 30.3 (29.2,31.5) |
| Large bag | 130 | 87.0 (74.4,94.2) | 80.3 (65.6,90.0) | 751 | 81.9 (77.0,86.0) | 76.6 (71.1,81.4) |
| No bag | 5,810 | 35.2 (33.3,37.1) | 14.5 (13.0,16.1) | 20,290 | 48.8 (47.7,49.9) | 29.4 (28.3,30.4) |
| Female | 3,500 | 41.8 (39.3,44.4) | 20.3 (18.1,22.7) | 11,944 | 53.0 (51.6,54.4) | 35.9 (34.5,37.4) |
| Male | 2,440 | 28.4 (25.6,31.2) | 9.9 (7.9,12.1) | 9,097 | 45.7 (44.1,47.3) | 25.1 (23.7,26.7) |

a; CI = confidence interval.

## Results

Table 2 contains the percentage avoidance of stairs (95% CIs) broken down by number of lifts, direction of travel and demographic group. Consistent effects of structural aspects of the building were found. Avoidance of stairs was more frequent when two lifts were available relative to one, and when going up relative to coming down. Superimposed on these structural influences were effects of demographics; avoidance of stairs was more common in the overweight, those carrying large bags and females than their comparison groups. As is clear from the non-overlapping CIs, effects of demographic group were present at each level of either structural aspect. The overweight avoided stairs more frequently than those of healthy weight, irrespective of lift availability or direction of travel. Presence of large bags was infrequent in the data set (3.3%), excluding it from formal analyses.

A preliminary omnibus analysis revealed more avoidance travelling up than down (OR = 3.22, 95% CI = 2.82, 3.67, $p < .001$). Analyses, summarised in Table 3, were performed separately for direction of travel.

As can be seen from the table, the overweight and females avoided stairs more than their comparison groups for both ascent and descent. Follow-up of the interactions between lift availability and weight status revealed more frequent avoidance of climbing with two lifts in healthy weight individuals (OR = 2.43, 95% CI = 2.18, 2.70, $p<0.001$) but no significant effect in the overweight (OR = 1.14, 95% CI = 0.99, 1.32, $p = 0.08$). For descent, two lifts were associated with a greater increase in avoidance for healthy weight (OR = 3.21, 95% CI = 2.75, 3.76, $p<0.001$) than overweight individuals (OR = 1.88, 95% CI = 1.54, 2.29, $p<0.001$). Additionally, there was more frequent avoidance with two lifts in males and females (all prob. $< .001$), with the ORs for males numerically greater than those for females in both directions.

**Table 3. Effects of demographics and lift availability on stair avoidance for ascent and descent in workplaces.**

| Variable | Up OR (95% CI)[a] | Down OR (95% CI) |
| --- | --- | --- |
| Overweight > Healthy weight | 2.43***[b] (2.09, 2.82) | 2.01*** (1.60, 2.52) |
| Females > Males | 1.90*** (1.62, 2.21) | 3.02*** (2.31, 3.95) |
| Two lifts > One lift | 2.96*** (2.54, 3.45) | 4.70*** (3.61, 6.14) |
| Weight status x lift interaction | 0.48*** (0.40, 0.57) | 0.59*** (0.46, 0.76) |
| Gender x lift interaction | 0.73*** (0.62, 0.87) | 0.59*** (0.45, 0.79) |

a; OR = odds ratio, CI = confidence interval.

b; * = p < .05, ** = p < .01, *** = p < .001.

## Discussion

In summary, the overweight, females and those carrying large bags avoided stairs more than their comparison groups. Overall, it appears that any tendencies for avoidance were enhanced by increases in the energy cost of the behaviour (up = 46.8%; down = 28.0%) and the availability of the method of avoidance (two lifts = 41.2%, one lift = 27.4%), with one exception. Avoidance of stair climbing by overweight individuals did not increase significantly when two lifts were available.

At work, avoidance of stair climbing was considerably less frequent than in public access settings, echoing research summarised in additional file, Table 1. Similarly, avoidance of stair descent with one lift (14.5%) was less frequent than when the alternative was an escalator (88.3%) in the same region of the UK [19]. Potential waiting time for a lift provides a plausible explanation for these differences in frequency of avoidance of lifts relative to escalators [6,13,22]. Additionally, pedestrians minimise the distance between destinations for level walking [23] and stair negotiation [19,24]. The layout of the building may influence choice. The stair exit could be closer to a photocopier on the left of the building whereas the lift exit may be closer to the office of a valued colleague on the right. Any distance minimization by an employee seeking a specific destination in the building could dilute effects of demographics. Further, stair choice at work is a frequently encountered option for most employees. Habitual behaviours develop when they are performed regularly in consistent contexts such as the workplace [25–27]. It seems likely that habits for stair and lift choice would develop for individual employees and might further dilute effects of demographics at each choice.

## Study 2

### Introduction

Occasionally in public access settings, a staircase is paired with a lift rather than an escalator. At such a site, the transition between levels is simply part of the journey. The location of both the top and bottom of either the stairs or the lift is essentially the same, a pause on the way to a range of possible destinations encircling the exit. Distance minimization may have minor effects on choice. Additionally, public access stairs will be encountered less frequently than those at work and habitual choice less likely to develop. The second study used a choice between stairs and a lift that connected pavements on different levels to test for the generality of effects of demographics when a lift was available outside of work. Based on study 1, demographic differences were predicted for both directions of travel and less avoidance was predicted for the lower energetic cost of descending.

### Methods

Observations of pedestrians ascending (n = 4,100) and descending (n = 3,333) a 43-step staircase (8.60m high) wrapped around a single central lift at an outdoor city centre site were made on eight days between 12:30 and 4:00pm. The entrance to lift was clearly visible to pedestrians at the top of the site but partially concealed from pedestrians approaching from the bottom unless a preceding pedestrian chose it. Trained observers situated at the top of the site coded pedestrians using the same criteria as the workplace data. One observer coded ascent and one descent within any time period. The Kappas were excellent for method of ascent (κ = 0.98), weight status (κ = 0.87), presence of large bags (κ = 0.92) and sex (κ = 0.96). Bootstrapped analyses (1000 samples) employed logistic regression with stair/escalator choice as the dichotomous dependent variable and the potential predictor variables of direction of travel, sex, presence of large bags and weight status.

**Table 4. Percentage avoidance of stairs (95% CI) outdoors, separated by direction of travel and demographic group.**

| Demographic | Total n = | One lift | |
|---|---|---|---|
| | | **Up** | **Down** |
| Overweight | 1,562 | 45.5 (41.7, 49.3) | 36.9 (32.7, 41.1) |
| Healthy weight | 5,871 | 22.3 (20.6, 23.9) | 17.0 (14.7, 17.9) |
| OR[a] (95% CI) | | 3.59***[b] (3.01, 4.27) | 3.67*** (2.99, 4.49) |
| Large bag | 669 | 48.9 (43.6, 54.1) | 53.7 (45.4, 61.4) |
| No Bag | 6,764 | 24.5 (22.9, 26.1) | 18.3 (16.8, 19.9) |
| OR (95% CI) | | 3.69*** (2.97, 4.60) | 7.54*** (5.50, 10.28) |
| Female | 3,099 | 39.1 (36.4, 41.8) | 27.9 (25.2, 30.6) |
| Male | 4,334 | 19.1 (17.3, 20.9) | 14.9 (13.1, 16.8) |
| OR (95% CI) | | 2.38*** (2.05, 2.77) | 1.93*** (1.60, 2.30) |

a; OR = odds ratio, CI = confidence interval for effects of that demographic.

b; * = p < .05, ** = p < .01, *** = p < .001.

## Results

A preliminary omnibus analysis revealed the expected main effect for direction of travel (OR = 1.29, 95% CI = 1.06, 1.56, *p* = .02). Table 4 contains the percentage avoidance of stairs (95% CIs) broken down by direction of travel and demographic group. The ORs (95% CIs) for the effects of demographic group for each direction are included in the table. Large bags were carried in 9.0% of the pedestrians and included in formal analyses.

As at work, stair avoidance was more common for ascent (27.3%, 95% CI = 25.7, 28.9) than descent (20.5%, 95% CI = 18.9, 22.1). Once again, the overweight, those carrying large bags and females avoided stairs more than their comparison groups, with effects superimposed on effects of direction of travel. Confidence intervals for demographic subgroups did not overlap in either direction. For weight status, similar differential rates of avoidance between healthy weight and overweight pedestrians occurred for ascent and descent.

## Discussion

Avoidance at this outdoor site echoed that found at work; effects of demographic group were superimposed on more frequent avoidance of ascent than descent. The general pattern of effects at work does not appear to be confined to a work environment where habitual choice might develop. Nonetheless, there were some differences between the two tested contexts. The frequency of avoidance of climbing increases with increasing height of the climb at work, consistent with the increased energetic cost [12,13,18]. The climb to the next level for the city centre site, 8.60m, was considerably higher than the average in the workplace data for the next floor, 3.45m, yet there was less frequent avoidance for the greater ascent (27.3%) than for one lift at work (35.2%). This less frequent avoidance could reflect either a lift that was partially concealed from pedestrians using the site or journeys of more than one floor in the workplace data. For descent, the lift was clearly visible at the top of the climb and the pattern expected from the difference in height appeared; there was *more* avoidance of descent at the higher outdoor site (20.5%) than at work (14.5%).

Fig 1 summarises the percentage avoidance of stairs for ascent and descent in healthy weight and overweight pedestrians in both studies. The overweight consistently avoided stairs more than those of healthy weight. The difference between the subgroups was of smaller magnitude when two lifts were available. For ascent, increased avoidance by healthy weight

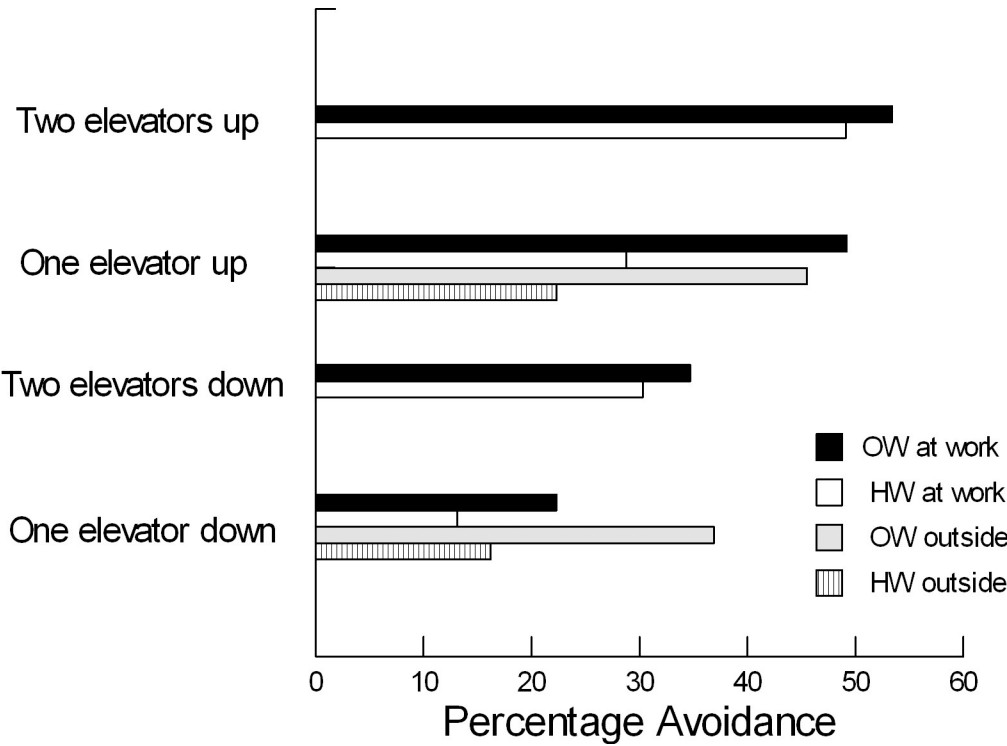

**Fig 1. Avoidance of stairs for ascent and descent separated by weight status and number of lifts (OW = overweight, HW = healthy weight).**

individuals with two lifts narrowed the gap between subgroups; changes for the overweight subgroup were not statistically significant. For the single lift at the outdoor site, the overweight avoided descent considerably more frequently than healthy weight pedestrians, with the greater magnitude difference than seen at work consistent with differences in energetic cost between sites. One simple fact would summarise these data and the earlier public access studies [9]. There was a generalised propensity for the overweight to avoid expending energy on stairs when a mechanised alternative was available.

## Study 3

### Introduction

In modern cities, sloped ramps can be provided as an alternative to stairs in some public access settings. This provision allows equal access for wheelchair users who cannot negotiate stairs. The final study investigated stair versus slope choice, and the associated demographics, at two such sites to provide replicated observations for the unusual outcome. Stair climbing requires up to three times the peak forces at the knee as level walking [28]. The more gradual climbing available on slopes requires less flexion of the knee and, hence, allows more gradual force production [29].

Chamberlain Square in Birmingham (UK) features a set of stairs opposite the entrance to the square and a sloping ramp around the periphery of the square that reaches the same destination as the top of the stairs (see S3 File for a picture of the site). Pedestrians at this site can avoid the stairs by walking up the slope. Preliminary observations at this site using standardised criteria [20] revealed that 41.2% (95% CI = 40.6, 42.0) avoided the stairs, with avoidance more common in females and those carrying large bags (n = 14,513; Eves, unpublished). The

second site was at one exit from the University of Birmingham campus towards University station (see S3 File). A short flight of stairs on the direct route to the station contrasted with a slope that allowed avoidance of the stairs.

## Methods

Observations of pedestrians travelling to the exit from Chamberlain Square (n = 13,768; 13 days) and towards the university station (n = 2,529; eight days) were made between 9:50am and 4:00pm. Only ascending pedestrians were coded. In Chamberlain Square, trained observers at the exit of the site coded pedestrians using the same criteria as the workplace data. One observer coded stair users and one coded slope users within any time period. For University station, a single observer alternated between stair and slope coding each 30 minutes. Double coding revealed excellent Kappas for method of ascent ($\kappa$ = 0.98), weight status ($\kappa$ = 0.84), presence of large bags ($\kappa$ = 0.80) and sex ($\kappa$ = 0.98). Bootstrapped analyses (1000 samples) employed logistic regression with stair/slope choice as the dichotomous dependent variable and the potential predictor variables of sex and weight status. Pedestrians carrying large bags were infrequent in both settings and excluded from formal analyses (Chamberlain Square = 1.8%; University = 1.7%).

## Results

Overall, avoidance of the stairs was less frequent in Chamberlain Square, 47.3% (95% CI = 46.4, 48.3), than when approaching University station, 65.2% (95% CI = 63.0, 67.3). Table 5 contains the percentage avoidance (95% CI) broken down by demographic group at both sites, with the ORs (95% CIs) for the effects of demographic group.

At both sites, the overweight and females avoided stairs more than their comparison groups. Confidence intervals for demographic subgroups did not overlap at either site. The effects of sex were of greater magnitude at University station than at Chamberlain Square.

## Discussion

As with lifts, avoidance of stairs by choosing a sloped ascent was frequent. The lower rates of avoidance in Chamberlain Square than the station may reflect the greater detour in the square

**Table 5. Percentage avoidance of stairs (95% CI) by choosing slopes at Chamberlain Square and University station by demographic group.**

| Demographic | Total n = | Chamberlain Square | Total n = | University Station |
|---|---|---|---|---|
| Overweight | 1,860 | 70.1 (67.6, 72.4) | 244 | 79.9 (73.3, 85.2) |
| Healthy weight | 11,908 | 43.8 (42.8, 44.8) | 2,285 | 63.6 (61.3, 65.8) |
| OR (95% CI)[a] | | 2.93***[b] (2.63, 3.25) | | 2.22*** (1.60, 3.07) |
| Large bag | 245 | 76.7 (69.9, 82.4) | 48 | 85.4 (69.2, 94.3) |
| No bag | 13,523 | 46.8 (45.9, 47.8) | 2,481 | 64.8 (62.6, 66.9) |
| OR (95% CI) [c] | | - | | - |
| Female | 6,074 | 51.2 (49.7, 52.6) | 1,308 | 70.8 (67.9, 73.6) |
| Male | 7,694 | 44.3 (43.1, 45.6) | 1,221 | 59.1 (55.9, 62.3) |
| OR (95% CI) | | 1.24*** (1.16, 1.33) | | 1.65*** (1.40, 1.95) |

a; OR = odds ratio, CI = confidence interval for each effect of demographic group.

b; * = p < .05, ** = p < .01, *** = p < .001.

c; insufficient data for inclusion in the analyses.

to choose the slope. Avoidance in the square required 76m of walking versus 48.4m for the direct route across the square; at the station, the discrepancy was smaller, 46.1m versus 42.6m. Inevitably, any detour would increase journey times and, typically, pedestrians seek to minimise time and distance [14,15,23,24]. Less frequent avoidance in the square may reflect the greater temporal cost of the indirect route. Nonetheless, stairs are a more energy efficient means of raising weight against gravity than slopes of equivalent angle [29]. Avoiding stairs by choosing a slope will *increase* both temporal and energetic costs, unlike the reduced costs with mechanised alternatives. This result of avoidance, despite increased cost, may reflect the more gradual force production possible on the slope to achieve the ascent; the actual height of the climb was the same for both alternatives.

## General discussion

The studies in this paper reveal consistent effects of demographic grouping on stair avoidance when an alternative is available. In workplaces, and at an outdoor site where the alternative was a lift, overweight pedestrians, those carrying large bags and females were more likely to avoid stairs than their comparison groups. In the final study, this pattern of avoidance occurred where the alternative method of ascent was a sloped ramp. Taken together with a previous summary of avoidance with escalators [9], these studies expand on the original question posed by Brownell and co-workers about the effects of weight status on avoidance of stair use [7,8]. Overweight pedestrians negotiating the built environment *are* more likely to avoid the physical activity of stair use as part of daily life than healthy weight pedestrians. So are females and those carrying large bags. A bias to minimise the costs of active transport provides a plausible explanation for this generality.

### Minimizing energetic cost

During locomotion, humans naturally optimise energetic cost. They adopt a step width, step length and step frequency for walking and choose a step length and frequency for running, all of which minimise the total metabolic cost for completion of the behaviour (see [30]). Humans have an optimal speed for walking and running that minimises the energetic cost per unit distance [31,32], as do other animals [33,34]. Minimisation of transport costs may be a universal default. This minimization requires repeated iterations to optimise the behaviour. Minimisation of transport costs is learnt, linked to the changes in the visual consequences of forward motion [30,35–38]. All of the above studies were for locomotion on the level. Stair climbing, at two and half times the energetic cost of purposeful walking [2], is a metabolically costly barrier encountered during active transport. A consistent bias for pedestrians to minimise the cost incurred by climbing is evident; in shopping malls where journey time is less of an issue than in stations, 92.4% avoid stairs (n = 355,069 [9]). Raising body weight against gravity is energetically costly and appears to be minimised by other animals [39,40]. Energy expenditure serves three main functions, basal metabolic rate, diet-related expenditure on ingested food, and energy for physical activity [41]. Human basal metabolic rate requires 60% of the recommended daily intake and utilizing food a further 6–12% [41]. At least two thirds of recommended intake are required for these recurrent costs of maintaining function that must be met. The only modifiable part of the equation linking intake and expenditure is the remaining third of intake available for movements of the body; it has been estimated that 89% of these movements involve walking [41]. Transport-related walking has deep evolutionary roots. Minimising the proportion of total intake required for transport would be biologically advantageous [30–34].

Nonetheless, the final pair of studies demonstrated that avoidance was not synonymous with minimisation of expenditure. Choosing a sloped ascent increased both temporal and energetic costs. Walking up a slope allows a more gradual force production during the ascent; peak forces at the knee are reduced compared to stairs [28,29]. Similarly, a modified climbing gait in older individuals reduces the forces at the knee and the ankle to a lower proportion of their maximal capacity [42]. Older climbers have reduced resources for climbing and adopt a reduced, and more gradual, increase in force over time for each step [42]. Choosing the slope in the final pair of studies would allow individuals with reduced resources for climbing to maintain output at a lower proportion of total resources, despite increases in energetic and temporal costs. Aggregated avoidance with ramped ascent, 50.1% (95% CI = 49.3, 50.9) exceeded that when the lift was the alternative, 33.5% (95% CI = 33.0, 34.0). Preference for more gradual resource expenditure could increase avoidance when a ramp was available whereas unwillingness to wait for a lift could decrease it.

## Perception of stair slope

Recent research on the perception of the slope of stairs provides clues to the mechanisms that may underlie avoidance based on climbing resources. Perception of the angle of hills and stairs is exaggerated in explicit awareness; a 10˚ hill is reported to be about 30˚ and a 23˚ staircase reported to be 45˚ [43–45]. In experimental studies, fatigue from an exhausting run [45–47], carrying extra weight [46] and depleted glucose resources [48], all result in further exaggerations of reported angle. While effects of experimental demand have been proposed as an alternative explanation [49,50], quasi-experimental studies confirmed effects of depleted resources [51] and additional weight carried [43,52] where demand was absent. Travelers waiting for their trains were recruited to complete an interview about the environment [43,51,52]; there was no experiment. Proffitt argued that perception of slope was 'embodied' in that resources for climbing influenced explicit perception [44]. Embodied effects of resources facilitate physical activity choices without individuals having to specifically consider resource availability [44].

Echoing the behavioural differences documented here, overweight pedestrians, those wearing heavy bags and females all reported potential climbs as steeper than their comparison groups [43,45–47,51,52]. Estimates of stair steepness scaled by the deadweight of fat mass that would be carried [51]. As noted earlier, females have, on average, a greater percentage of their weight as body fat (25%) than males (12.5%) and hence are encumbered by more deadweight [10]. Consistently, females reported slopes as steeper than males did [43,45,47,49,50]. Further, stairs were reported as steeper by pedestrians who avoided them by choosing the escalator, even when potential effects of demographics were controlled by stratified sampling and statistical adjustment [43]. Perceived steepness appears to be an environmental cue linked to resources that can deter climbing when an alternative is available [43].

At work, the stairs may not be directly visible when a lift is chosen so perception of steepness cannot directly influence choice. Nonetheless, individuals learn about the potential cost of climbing from experience, biasing subsequent choices. As resources change, behaviour, and the associated perception, echo these changes. Body mass is composed of fat free mass and fat mass. Fat free mass that provides resources for climbing was unrelated to perceived steepness [52]. Rather, it was fat mass, i.e. deadweight that must be carried, that was linked to perception [52]. Further, only changes in fat mass, not fat free mass, were related to changes in perception of steepness over a year later [52]. Similar calibration of perception occurred for changes in body size during pregnancy [53] and loss of leg strength with ageing [54]; changes without experience were ineffective [53]. Obese individuals walk less than lean participants, with daily

walking distance negatively correlated with body fat [55]. In the only truly experimental study of weight change, increases in body mass, 78% of which was fat, reduced the distance walked equally for healthy and overweight participants [55]; resources changed behaviour. In both longitudinal perception studies, body change over time influenced perception [52,53]. No study has altered perception to change behaviour. The simplest conclusion is that weight carried deters behaviour [55], consistent with the direction of effects in other research [56–58]. Avoidance of stairs seems likely to be a consequence of weight carried. The role of learning in this process, and the potential mediator of perception, is unknown.

## Limitations

It is a curiosity that one strength of these data, direct auditing of behaviour, is accompanied by a limitation. Auditing provides matchless accuracy about the actual behaviour performed; accelerometers, for example, cannot identify behaviour. Observational studies of stair use allow a test of the biasing effects of extra weight carried because of the ability to clearly identify the behavioural choice made. Stair climbing is a vigorous member of the family of active transport behaviours. The energetic cost of stair climbing is clear. Work done to raise weight against gravity is relatively independent of the rate of climbing. Height of the climb, not speed, primarily determines cost. Climbing at 60 steps.min$^{-1}$ required 8.7 METs (Eves & White, unpublished) whereas climbing at almost twice that speed, 110 steps.min$^{-1}$, cost 9.6 METs [2]. Effects of weight on the lower intensity activity of stair descent here, and on 'objectively' measured walking [55], physical activity [57] and sitting time [56], indicate a generalised effect of weight carried on physical activity choices. Nonetheless, auditing will imperfectly measure demographic differences. Sex is generally straightforward but weight status, and the additional weight of a large bag, must be imprecise categories, even when silhouettes optimise coding for weight status [3,21]. The commonality of effects of weight carried on avoidance across different settings, however, does not suggest imprecision in measurement has impeded the research. The fact that demographic differences in avoidance behaviour are linked to perception of an environmental cue that promotes avoidance indicates some triangulation on the question.

## Conclusions

Overweight pedestrians, those carrying large bags and females were more likely to avoid stairs when a lift was available at work and outdoors, as well as when the alternative to stairs was a sloped ascent. Weight carried can be an encumbrance that deters participation in the incidental physical activity of stair usage during daily life. Increased physical activity, including stair climbing, is a current target of public health in the developed world to improve population health [1,59–61]. A focus on environments that would encourage active transport such as walking is prominent in current approaches [62–64]. Mechanisms that minimise the costs of active transport can run counter to public health initiatives to increase physical activity as part of daily life.

## Supporting information

**S1 File. Previous studies of the effects of demographic grouping on stair avoidance at work.**
(PDF)

**S2 File. Building characteristics for study 1.**
(PDF)

**S3 File. Chamberlain Square and University Station.**
(PDF)

## Acknowledgments

I thank Mike White, Janice Thompson and Doug Carroll for helpful comments on earlier versions of this paper, and the statistician, Roger Holder, for the recommendation to use bootstrapped analyses. I thank Birmingham and Coventry City Councils, Severn Trent, Pilkingtons and Birmingham Chamber of Commerce for permission to use their buildings as well as Birmingham City council for permission to make observations at the outdoor site and in Chamberlain Square. Finally, I thank the third-year project students without whom these observations would not have been collected.

## Author Contributions

**Conceptualization:** Frank F. Eves.

**Data curation:** Frank F. Eves.

**Formal analysis:** Frank F. Eves.

**Funding acquisition:** Frank F. Eves.

**Investigation:** Frank F. Eves.

**Methodology:** Frank F. Eves.

**Project administration:** Frank F. Eves.

**Resources:** Frank F. Eves.

**Software:** Frank F. Eves.

**Supervision:** Frank F. Eves.

**Validation:** Frank F. Eves.

**Visualization:** Frank F. Eves.

**Writing – original draft:** Frank F. Eves.

**Writing – review & editing:** Frank F. Eves.

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
