## [Decision Letter · Decision Letter 0]

20 Nov 2019

PONE-D-19-27601

When weight is an encumbrance; avoidance of stairs by different demographic groups.

PLOS ONE

Thank you for submitting your manuscript to PLOS ONE. After careful consideration, we feel that it has merit but does not fully meet PLOS ONE’s publication criteria as it currently stands. Therefore, we invite you to submit a revised version of the manuscript that addresses the points raised during the review process.

This is a fine paper, however, one reviewer has requested minor revisions we would like you to address before acceptance.

We would appreciate receiving your revised manuscript by Dec 26 2019 11:59PM. To enhance the reproducibility of your results, we recommend that if applicable you deposit your laboratory protocols in protocols.io, where a protocol can be assigned its own identifier (DOI) such that it can be cited independently in the future. For instructions see: http://journals.plos.org/plosone/s/submission-guidelines#loc-laboratory-protocols

We look forward to receiving your revised manuscript.

Kind regards,

Lee Smith

Academic Editor

PLOS ONE

Journal Requirements:

2. Please reorganise your manuscript and incorporate the sections of each separate study into a unique section. For more information please visit https://journals.plos.org/plosone/s/submission-guidelines .

4. Please upload a copy of Figure 2, to which you refer in your text on page 17. If the figure is no longer to be included as part of the submission please remove all reference to it within the text.

Reviewers' comments:

Reviewer's Responses to Questions

**Comments to the Author**

1. Is the manuscript technically sound, and do the data support the conclusions?

Reviewer #1: Yes

Reviewer #2: Yes

2. Has the statistical analysis been performed appropriately and rigorously? 

Reviewer #1: Yes

Reviewer #2: Yes

3. Have the authors made all data underlying the findings in their manuscript fully available?

Reviewer #1: Yes

Reviewer #2: Yes

4. Is the manuscript presented in an intelligible fashion and written in standard English?

Reviewer #1: Yes

Reviewer #2: Yes

5. Review Comments to the Author

Reviewer #1: This paper reports a literature review, and three new empirical studies, that together demonstrate that those for whom stair use is most physiologically costly are most likely to avoid using the stairs. The paper is well-written and the studies use a rigorous methodology, including direct observation of thousands of participants in real-world settings. The paper furthers understanding of reasons for stair use, or rather avoidance thereof, and has important implications for public health policy and practice. I have only very minor suggestions for improvement.

Abstract: Please state explicitly the number of studies presented in the manuscript.

Abstract, Discussion subsection: ‘minimization of physiological costs of transport-related walking biases behaviour towards avoidance of stairs when an alternative is available’. This statement seems an over-generalisation; presumably it is not just any ‘alternative’ that must be available, but rather a lower-cost alternative.

Abstract and throughout paper: I found the term ‘lifestyle physical activity’ to describe stair use slightly awkward. While an accurate depiction, applying the term ‘lifestyle’ to the mundane act of climbing stairs has the effect (to me, at least) of inflating its importance and meaning to individuals. Might this be changed to ‘incidental physical activity’ or ‘instrumental physical activity’ instead?

Abstract, Discussion subsection: Both here, and in the Conclusion, the author argues that observed striving to minimize costs ‘runs counter to public health initiatives’. Which initiatives, and how does cost minimisation conflict with such initiatives? This point could be expanded on in the General Discussion.

Introduction, p4: The sentence beginning ‘For body weight, all physical activity contributes…’ is difficult to comprehend. Please simplify.

Introduction, p4: What is meant by the term ‘in the field’ in the first paragraph? Can this be deleted?

Preliminary study: This study summarises a ‘preliminary literature review’, but the methods used to conduct the review are not reported. Assuming this review was carried out for the purpose of this paper, the sources within the literature review should be explicitly stated, together with an indication of the process of identifying these sources.

Preliminary study, p6: The author reports here that while ‘nine studies reported that females avoided stairs more than males, five reported the opposite disparity’. Why should the opposite disparity have been found? This seems at odds with the arguments forwarded in the Introduction regarding groups most likely to avoid stair use, so should be considered in more depth.

Study 1, p11: Replace the term ‘no significant effect’ with ‘no effect’.

Reviewer #2: Excellent paper, very minor comments included.

Abstract: "This minimization of costs runs counter to public health initiatives that targets weight gain." Please specify what "this minimization of costs" strands for.

Gender and sex: throughout the paper, please check for consistency and use sex (male/female) or gender (men and women) or clearly define the different use of each. Currently sex and gender seem to be used interchangeably.

Method: pleaae define "large bags"

6. PLOS authors have the option to publish the peer review history of their article (what does this mean?). If published, this will include your full peer review and any attached files.

Reviewer #1: Yes: Benjamin Gardner

Reviewer #2: No

---

## [Author Response · Author response to Decision Letter 0]

5 Dec 2019

I thank the reviewers, and the editor, for their generally positive comments. I have responded to each of query below.

Reviewer #1: This paper reports a literature review, and three new empirical studies, that together demonstrate that those for whom stair use is most physiologically costly are most likely to avoid using the stairs. The paper is well-written and the studies use a rigorous methodology, including direct observation of thousands of participants in real-world settings. The paper furthers understanding of reasons for stair use, or rather avoidance thereof, and has important implications for public health policy and practice. I have only very minor suggestions for improvement.

I thank the reviewer for their kind comments.

Abstract: Please state explicitly the number of studies presented in the manuscript.

Now included in the abstract as requested.

Abstract, Discussion subsection: ‘minimization of physiological costs of transport-related walking biases behaviour towards avoidance of stairs when an alternative is available’. This statement seems an over-generalisation; presumably it is not just any ‘alternative’ that must be available, but rather a lower-cost alternative.

I can understand why the reviewer queries this statement. While the first two studies entail minimization of the energetic costs of active transport, choosing a ramp to ascend increases both energetic and temporal costs at both locations in the final study. In this case, it is likely that rate of expenditure is the key variable as I explain in the associated discussion section.

Abstract and throughout paper: I found the term ‘lifestyle physical activity’ to describe stair use slightly awkward. While an accurate depiction, applying the term ‘lifestyle’ to the mundane act of climbing stairs has the effect (to me, at least) of inflating its importance and meaning to individuals. Might this be changed to ‘incidental physical activity’ or ‘instrumental physical activity’ instead?

I understand the reviewer’s concern here and have changed the usage to either ‘the lifestyle physical activity of active transport’ or ‘incidental physical activity of stair climbing/usage as part of daily life’.

The paper reports a bias to avoid stairs as a result of extra weight carried. This is true for both climbing and the less intense activity of stair descent. Such a bias is also seen for the only experimental data for walking on the level (Levine et al., 2008) and the epidemiological studies of Metcalf et al., (2011) and Ekelund et al., (2008) that demonstrate that extra weight reduces what are termed ‘objective measures’ of physical activity. Observational studies of stair use allow a test of the biasing effects of extra weight carried because of the ability to clearly identify the behavioural choice made. I think that these effects generalise and have made this point explicitly in the discussion section (page 22, line 289–292, page 23 line 295-298). I thank the reviewer for the greater clarity about my underlying assumptions and the implications of the data resulting from this requested clarification.

Abstract, Discussion subsection: Both here, and in the Conclusion, the author argues that observed striving to minimize costs ‘runs counter to public health initiatives’. Which initiatives, and how does cost minimisation conflict with such initiatives? This point could be expanded on in the General Discussion.

I apologise for the oversight. I now make explicit in the general discussion that increased physical activity is a current target of public health in the developed world and provide appropriate references (page 23, line 311-314).

Introduction, p4: The sentence beginning ‘For body weight, all physical activity contributes…’ is difficult to comprehend. Please simplify.

I have reworded the sentence to read ‘For body weight, all physical activity requires energy expenditure that counters intake that can lead to weight gain.’ I hope this is now clearer.

Introduction, p4: What is meant by the term ‘in the field’ in the first paragraph? Can this be deleted?

I have removed the phrase and replaced it with ‘outside the laboratory’ to make explicit that this was not a laboratory study but rather one that used an 11-story building.

Preliminary study: This study summarises a ‘preliminary literature review’, but the methods used to conduct the review are not reported. Assuming this review was carried out for the purpose of this paper, the sources within the literature review should be explicitly stated, together with an indication of the process of identifying these sources.

The information about the process and sources for the preliminary literature review are presented in the supplementary file S1.

Preliminary study, p6: The author reports here that while ‘nine studies reported that females avoided stairs more than males, five reported the opposite disparity’. Why should the opposite disparity have been found? This seems at odds with the arguments forwarded in the Introduction regarding groups most likely to avoid stair use, so should be considered in more depth.

I agree with the reviewer that greater avoidance in men is puzzling. I have no clear explanation but have inserted the sentence. ‘The anomalies of greater avoidance in men might reflect differing locations by gender of meaningful journey ends within the building.’ (page 6, line 137ff).

Study 1, p11: Replace the term ‘no significant effect’ with ‘no effect’.

I have resisted this suggestion. The OR of 1.14, (95% CI = 0.99, 1.32, p=0.08) is close to significance and I feel that the unqualified ‘no effect’ would be misleading.

Reviewer #2: Excellent paper, very minor comments included.

I thank the reviewer for their kind comment.

Abstract: "This minimization of costs runs counter to public health initiatives that targets weight gain." Please specify what "this minimization of costs" strands for.

The sentence has been altered to now read ‘This minimisation of physical activity costs runs counter to public health initiatives to increase activity to improve population health.’

Gender and sex: throughout the paper, please check for consistency and use sex (male/female) or gender (men and women) or clearly define the different use of each. Currently sex and gender seem to be used interchangeably.

I thank the reviewer for their vigilance. I have amended the manuscript to make explicit that I am referring to sex.

Method: pleaae define "large bags"

I now make explicit that ‘large bags were sizes greater than a briefcase or small rucksack.’

---

## [Editor Report · Decision Letter 1]

7 Jan 2020

When weight is an encumbrance; avoidance of stairs by different demographic groups.

We are pleased to inform you that your manuscript has been judged scientifically suitable for publication and will be formally accepted for publication once it complies with all outstanding technical requirements.

With kind regards,

Lee Smith

Academic Editor

PLOS ONE

Additional Editor Comments (optional):

Thank you for adequately responding to the reviewer comments. I have now recommended that this paper is accepted.
---

## [Editor Report · Acceptance letter]

8 Jan 2020

PONE-D-19-27601R1 

When weight is an encumbrance; avoidance of stairs by different demographic groups. 

Dear Dr. Eves:

I am pleased to inform you that your manuscript has been deemed suitable for publication in PLOS ONE. Congratulations! Your manuscript is now with our production department. 

With kind regards,

on behalf of

Dr. Lee Smith 

Academic Editor

PLOS ONE